# Transparent number-naming system gives only limited advantage for preschooler's numerical development: Comparisons of Vietnamese and French-speaking children

**Mai-Liên T. Lê**[1,2]*, **Marie-Pascale Noël**[1]

**1** Psychological Sciences Research Institute, Université Catholique de Louvain, Louvain-la-Neuve, Belgium, **2** Faculty of Psychology, University of Social Sciences and Humanities, National University of Vietnam in Ho Chi Minh City, Ho Chi Minh City, Vietnam

☯ These authors contributed equally to this work.
* thi.m.le@uclouvain.be

## Abstract

Several cross-sectional studies have suggested that the transparency of the number-naming system of East Asian languages (Chinese, Japanese) facilitates children's numerical development. The Vietnamese number-naming system also makes the base-10 system very explicit (eleven is *"mười một,"* literally "ten-one," and thirty is *"ba mươi,"* literally "three-ten"). In contrast, Western languages (English, French) include teen words (eleven to sixteen) and ten words (twenty to ninety) that make their counting systems less transparent. The main question addressed in this paper is: *To what extent does a language's number-naming system impact preschoolers' numerical development*? Our study participants comprised 104 Vietnamese and 104 French-speaking Belgian children between 3½ and 5½ years of age, as well as their parents. We tested the children on eight numerical tasks (counting, advanced counting, enumeration, Give-N, number-word comparison, collection comparison, addition, and approximate addition) and some general cognitive abilities (IQ and phonological loop by letter span). The parents completed a questionnaire on the frequency with which they stimulated their child's numeracy and literacy at home. The results indicated that Vietnamese children outperformed Belgian children only in counting. However, neither group differed in other symbolic or non-symbolic abilities, although Vietnamese parents tended to stimulate their child at home slightly more than Belgian parents. We concluded that the Vietnamese number-naming system's transparency led to faster acquisition of basic counting for preschoolers but did not support other more advanced numerical skills or non-symbolic numerical abilities. In addition, we extended the evidence that both transparent number-naming system and home numeracy influence young children's counting development.

**Data Availability Statement:** All relevant data are within the manuscript and its Supporting Information files.

**Funding:** In this study, the first author was supported by the Belgian Academy for Research and Higher Education (ARES-Belgium), and the second author by the Fund for Scientific Research of the French-Speaking Community of Belgium (FNRS); https://www.ares-ac.be; https://www.frs-fnrs.be/fr/. The funders had no role in study design, data collection, and analysis, decision to publish, or preparation of the manuscript.

**Competing interests:** The authors have declared that no competing interests exist.

# Introduction

Cross-national studies have shown better mathematics performance in schoolchildren from Asian countries, such as Singapore, China, Hong Kong, Taiwan, Japan, South Korea, and Vietnam [1]. Previous studies have tested children from East Asia, mostly from China [2–7] (and more rarely, Taiwan, Korea, or Japan) [8, 9]. They found that Chinese children tend to outperform Western children in terms of different numerical abilities at preschool age (see [2] for a review) such as counting [3–5], counting large sets of objects [3, 6], calculation [7], and non-symbolic comparison [6]. However, Chinese culture and education are quite different from their Western counterparts in many ways (see [10, 11] for a review). Thus, the differences observed between Asian and Western children are possibly due to the differences in the number-naming systems of their respective languages, the learning of number concepts at school [12], and home education differences [13–16].

This introduction begins with a brief overview of young children's numerical development. We summarize the findings regarding how this development is affected by language and home numeracy differences and report the research that has tried to disentangle these two effects. Finally, we present our outcomes.

## Overview of numerical development

At birth, babies already show an ability to roughly perceive the numerical magnitude of a set of items due to their approximate number system (ANS) [17]. This inherited, non-symbolic magnitude system entails subitizing and approximation abilities. For instance, a child estimates and discriminates large quantities quickly and intuitively without counting [18]. Later, children move on to learning how to count and progressively master counting strings [19]. More advanced counting skills develop in parallel, such as reciting a number sequence from a number other than one [19]. At first, children recite a counting sequence without knowing its meaning. Progressively, they use a counting sequence while pointing at each object of a set (i.e., procedural knowledge of enumeration) and progressively grasp that the last number word counted refers to the cardinal of the set (i.e., the cardinal principle, [20, 21]). Later, children can perform the "*Give-N*" objects task, showing that they understand the cardinal meaning of the number "*N*" (i.e., conceptual knowledge of cardinality) [22]. Wynn showed that the acquisition of number words' cardinal meaning does not coincide with the abilities of sequence counting or enumerating objects [22]. For example, in one study, many 3-year-olds who were able to count to "*five*" failed when requested to give a set of five objects. Once children figure out the cardinal principle for smaller numbers (up to three or four), they generalize this principle to the rest of their counting list [21, 22]. Understanding the cardinal meaning of number words serves as the basis for the magnitude comparison of number words and, later on, of Arabic numbers. Moreover, children with no training in formal arithmetic can approximate addition and subtraction [23]. They can estimate the sum of two Arabic numbers and compare it to a third number, independent of the knowledge of exact numbers and arithmetic instruction [24].

## Effect of language on the numerical development of preschoolers

Numerical development is influenced by different language characteristics related to the numerical lexicon (i.e., the regularity and transparency of number-naming systems), numerical morphology, or phonological and syntactic properties of language [25]. The effect of the transparency of the number-naming system has been the subject of many studies. Dowker, Bala, and Lloyd [26] defined "*the regularity (i.e., transparency) of the spoken number system by the degree to which it gives a clear and consistent representation of the base system (usually base*

*10) used in the language and the consistency of conformity between the spoken and the written number system (usually the Arabic number system)*" (p. 525). According to both of the above criteria, East Asian languages, such as Chinese, Japanese, and Korean, have very regular number-naming systems (e.g., 11 literally means "ten-one"). On the other hand, irregular number-naming systems, such as those of English or French, include number-naming words that do not show a one-to-one correspondence with the Arabic written system. For instance, they have teen and decade number words (e.g., "thirteen" and "twenty") that need to be learned separately.

In a classic study, Miller, Smith, Zhu, and Zhang [3] showed that 4- to 5-year-old Chinese children could generally count up to 40, while American children of the same age could barely get to 15, and it takes them another year to reach 40. Interestingly, there were no differences in counting performance to 10, but fewer American children than Chinese children could count to 20. Another study [4] assessed two counting skills, counting up to 20 and counting from 9 to 15, and found that 5-year-old Chinese children had better counting skills than British and Finnish children. Furthermore, 6-year-old Chinese children were better at simple addition than their American peers [7].

The majority of studies are concerned with the impact of the number-naming system on symbolic numerical ability (i.e., counting, enumeration, addition), but few studies have examined the effects of language on non-symbolic abilities. On average, adults in math-literate societies have slightly higher ANS acuity than adults from cultures that do not use systematic count lists, such as the Mundurukus (i.e., an Amazonian ethnic group with a very small lexicon of number words) [27]. Interestingly, some Mundurukus have gone to school. Piazza and colleagues [28] found that when they entered school and learned to count in Portuguese (a language with a repetitive, straightforward counting list from 1 to 19), their ANS acuity improved.

Based on these results, some authors have hypothesized that a transparent number-naming system (e.g., "ten-one" for 11), which may enhance Chinese children's learning of symbolic numbers [29], could also boost children's processing of non-symbolic numbers [5, 6, 30]. Accordingly, previous studies have showed that Chinese children (5- to 7-year-olds and 4- to 6-year-olds) tend to perform moderately better on a non-symbolic numerical task than British and German children, respectively [5, 6]. Similarly, Dowker and Roberts [30] demonstrated that Welsh children (in grade 2) who speak a language with the same number-word structure transparency as Asian languages performed better on non-verbal line estimations than their English peers.

Interestingly, regarding the understanding of number words' cardinal meaning, tested using the Give-N task, Asian children are delayed compared to Western children [31, 32]. For instance, 2- to 3-year-old Japanese and 2- to 4-year-old Chinese children lagged behind their English- and Russian-speaking peers in their understanding of the cardinal value of the number words "*one*," "*two*," and "*three*" [31]. Similarly, Le Corre, Li, Huang, Jia, and Carey [32] found that Chinese children aged between 2 and 3½ who speak Mandarin learned the meaning of the number word "*one*" three to six months later than English learners. These findings have been interpreted as the effect of the language's numerical morphology (i.e., singular/plural marking in a Western language such as English or Russian, but not in Asian languages such as Japanese or Mandarin) [31, 32]. This research shows that languages that distinguish between singular and plural support the understanding of number-word cardinal meaning (at least of the first number "*one*").

## Home numeracy and preschoolers' numerical performance

In addition to the difference in the degree of transparency of the number-naming system, other factors could account for the cross-cultural differences between Asian and Western

children, such as differences in schooling, overall exposure to mathematics [10–12], or parent-child stimulation at home [13–16]. Children in East Asian countries receive more numeracy practice and instruction than children in Europe and North America [12]. Notably, the frequency of parent-child numeracy activities is significantly related to young children's mathematical ability ([33, 34]; see also [35] for a review). LeFevre, Clarke, and Stringer [36] assessed 27 French- and 38 English-speaking preschoolers' numeracy skills. They found that parent-child numeracy frequency was directly related to both groups' counting abilities.

Skwarchuk, Sowinski, and Le Fevre [37] distinguished between formal and informal numeracy activities. They defined formal numeracy activities as "*shared experiences in which parents teach directly and intentionally to their children the numbers*, *quantity or arithmetic to improve numeracy knowledge"* and informal numeracy activities as "*shared activities where teaching about numbers*, *quantity or arithmetic is not the goal of the activity but can happen incidentally*" (p. 65). Informal numeracy activities include board games or seller's games. According to the literature, Chinese-American and Taiwanese-Chinese parents tend to offer more formal mathematics instruction at home, structure their children's time to a greater extent, and engage children in mathematics-related activities at earlier ages [13]. European-American parents focus on building conceptual knowledge using informal, incidental methods rather than emphasizing skill practice [10]. Chinese mothers teach their 5- to 7-year-old children more computation than American mothers [14].

Additionally, Skwarchuk, Sowinski, and LeFevre [37] found that formal home numeracy practices (e.g., practicing simple sums) predicted children's knowledge of symbolic number systems. By contrast, reports on informal exposure to games with numerical content (measured indirectly through parents' understanding of children's games) predicted children's non-symbolic arithmetic. Studies of Asian parents have shown that Chinese mothers' participation in number skill activities and fathers' involvement in number games and application activities significantly predicted their children's mathematical performance [16]. Similarly, Chinese-American parents' more frequent and diverse numeracy activities were associated with their children performing better on numeracy tasks than their European-American peers [13].

Numeracy-related experiences and language have often been studied separately as a potential explicator of cross-national variation in early numeracy. However, the effects of language and experience are often closely connected [38]. In many cases, research designs have not entirely disentangled the effects of language and home experiences. Accordingly, existing cross-national studies have predominantly focused on comparing language (e.g., English and Chinese) [3] or parents' practices (e.g., American and Chinese parental involvement) [14]. Only rarely have researchers explored multiple factors simultaneously or attempted to control or match groups on some variables ([35], for a review); for example, the role of the number naming system and parents' influence simultaneously [39].

## Present study

The present study aims to examine the impact of language (i.e., more specifically, the degree of base-10 transparency of the number-naming system) on preschoolers' numerical development.

Most previous studies focused on cross-national differences between children from East Asia (China, Korea, and Japan) and Western Europe (the UK and France). However, these two samples differ regarding the number-naming system structure and children's mathematical teaching and home numeracy stimulation. To differentiate between global cultural differences and specific linguistic effects, we chose to use another Asian country, Vietnam (a former

French colony), and compare it with Belgium's French-speaking region. Vietnam's rapid economic growth in the past 30 years has transformed it from one of the world's poorest countries to a middle-income country [40]. Moreover, Vietnamese society now has more access to Western cultures and beliefs [41] and is now more open to individualistic values [42]. Additionally, as the difference between children raised in Vietnam and Belgium might be related to other factors, we also examined the role of home numeracy experience in preschoolers' numerical development.

Vietnamese has a very regular, transparent base-10 number-naming system, whereas French does not. For instance, in Vietnamese, 11 is *mười một* ("ten-one"), 12 is *mười hai* ("ten-two"), 20 is *hai mươi* ("two-ten") 30 is *ba mươi* ("three-ten"), and 59 is *năm mươi chín* ("five-ten-nine"). The French number-naming system in Belgium is less transparent and less regular [43]; it includes ten words (11 to 16) and ten words (20, 30, etc., up to 90) that make the counting system less transparent than the base-10 structure. Furthermore, like Mandarin or Japanese, Vietnamese does not have a singular/plural distinction (see [44, 45] for a review), whereas this distinction is obligatory in French.

Most previous studies have tested basic counting in young children, but only a few tested advanced counting. Symbolic number comparison has only been assessed using Arabic numbers [5], while to the best of our knowledge, no study has investigated the magnitude comparison of spoken number words. In this study, we tested whether the transparency of the Vietnamese number-naming system facilitates the acquisition of basic counting and of other more advanced symbolic numerical abilities, such as the elaboration of number sequences (i.e., counting from a number; see [4]) and the enumeration of object sets (e.g., [3, 6]). We also used two tasks testing children's conceptual understanding of the number words' magnitude: the Give-N task and the number-word comparison task. A simple addition task was also included, as a difference in favor of Asian children has been identified previously [5, 7]. Furthermore, we wanted to examine whether these possible advantages of Asian children in symbolic number-processing tasks also extend to tasks that do not involve number words but simply tap into the magnitude processing of collections. Accordingly, we included a collection comparison task and an approximate addition task.

Based on the advantage of Asian languages' number systems for Asian children [3–10], we hypothesized that Vietnamese preschoolers would perform better on acquiring counting sequences, enumeration, and simple addition than their Belgian peers. For counting ability, we did not expect differences for numbers up to 10, but for larger numbers when number word structures differ. However, based on the lack of the singular/plural distinction in Vietnamese [31, 32], we expected that Vietnamese children's understanding of number words' cardinal value would be less than that of Belgian children. For now, due to a lack of evidence, the prediction regarding some symbolic abilities (i.e., verbal number comparisons, the elaboration of number sequences, and approximate addition) remains an open question. In terms of non-symbolic abilities, we could expect, based on previous studies [5, 6], that Vietnamese children may have a slightly better ANS acuity than Belgian children. Furthermore, based on previous studies [36, 38], we assumed that cross-cultural differences in home numeracy would also contribute to the differences between the two samples in the numerical tasks.

To test these hypotheses, we selected two samples of typical preschool children from Vietnam and Belgium. We chose children of preschool age to minimize school influences. We tested the children on their numerical abilities (counting, counting from a number, enumeration, the Give-N task, number-word comparison, collection comparison, addition, and approximate addition) and general cognitive ability (IQ and phonological loop). The parents completed a questionnaire on the frequency of numeracy and literacy activities at home with their child.

## Method

Ethics approval for this study was obtained from the Ethics Committee of Psychological Sciences Research Institute of the Catholic University of Louvain. Approval number: Projet2017-05. All children's parents were given written consent. The sample consisted of children whose parents consented to their children's participation.

### Participants

Children were recruited from six nursery kindergartens in Vietnam and four kindergartens in the French-speaking part of Belgium. To further minimize the cultural differences between the children from Belgium and the children from Vietnam, we selected our Vietnamese sample in Ho Chi Minh City (formerly known as Saigon) and Binh Duong, two developed cities in southern Vietnam. In Belgium, we chose French-speaking cities with an average socioeconomic level (Ottignies, Namur, and Charleroi).

The children came from middle-class socioeconomic backgrounds, and they were all monolingual Vietnamese speakers (VN) or Belgian French speakers (BEL). Three exclusion criteria were used: first, the children's information provided by the parents should not mention any neurodevelopmental disorders (e.g., autism) or developmental delays (4 VN & 10 BEL were excluded on this basis); second, the IQ score of the participants aged 4 years and older should not be below two SD from the mean (9 VN & 4 BEL were excluded on this basis); third, children who take extra mathematics courses outside of school were excluded from the study ($n$ = 23 VN). From a more extensive sample used in a longitudinal study of Vietnamese children (N = 310) and Belgian children (N = 151), we selected two groups of children that were as similar as possible in terms of gender, age, reasoning abilities, and parents' education level in the two countries.

The demographic data of the samples are shown in Table 1. The final sample was composed of 104 Vietnamese ($M$ = 54.8 months, $SD$ = 6.39; 54 girls and 50 boys) and 104 Belgian children ($M$ = 55.3 months, $SD$ = 6.98; 54 girls and 50 boys), ages between 42 and 66 months. The parents reported their highest education level. This information was then encoded into the number of schooling years because it differed slightly between Vietnam and Belgium (see Table 1). For instance, primary education is five years in Vietnam but six years in Belgium; moreover, the secondary education level in Vietnam includes seven years, while in Belgium, it is six years. University in Vietnam is for four years and independent of a master's degree (2 years), while in Belgium, university degrees are usually five years: a bachelor's degree (3 years) and a master's degree (2 years). For instance, for a parent who has completed primary and secondary education, the number of years of schooling is 12 in Belgium (6+6) and in Vietnam (5+7).

**Table 1. Distribution of the samples according to the education systems in Vietnam and Belgium.**

| Vietnamese Education System | | | | Belgian Education System | | | |
|---|---|---|---|---|---|---|---|
| Education Level | N Years of Schooling | N Mother | N Father | Education Level | N Years of Schooling | N Mother | N Father |
| 1. Primary | 5 | | | I. Primary | 6 | | 1 |
| 2. Lower secondary | 9 | 2 | 1 | II. Secondary | 12 | 19 | 29 |
| 3. Upper secondary | 12 | 10 | 10 | | | | |
| 4. Professional training | 14 | 13 | 11 | III. Superior | 15 | 43 | 27 |
| 5. College | 15 | 9 | 6 | | | | |
| 6. University | 16 | 65 | 67 | IV. University (bachelor's and master's) | 17 | 39 | 43 |
| 7. Master's | 18 | 5 | 8 | | | | |
| 8. Doctor of Philosophy | 22 | | 1 | V. Doctor of Philosophy | 21 | 3 | 4 |

Only a few Vietnamese parents had completed only primary and secondary education or professional training (25 mothers and 22 fathers), and a few Belgian parents had completed primary and secondary education (19 mothers and 30 fathers). Most parents had received a university (or higher) education (79 VN and 85 BEL mothers, 82 VN, and 74 BEL fathers). We were thus working with well-educated populations in both countries.

We introduced years of schooling for mothers and fathers in univariate ANOVAs by nationality (two levels: Vietnam and Belgium) as a between-subjects factor. There was no significant difference between the two samples in terms of the number of years of schooling, either for the mothers [$F(1, 206) = 0.318$, $p = 0.318$, $\eta^2 = 0.002$, $M = 15.24$, $SD = 1.65$ vs. $M = 15.38$, $SD = 2.02$] or for the fathers [$F(1, 206) = 1.85$, $p = 0.285$, $\eta^2 = 0.006$, $M = 15.45$, $SD = 1.65$ vs. $M = 15.13$, $SD = 2.52$]. Both mothers and fathers in each sample had an average of 15 years of schooling, which is equivalent to a college education (in Vietnam) or superior education (in Belgium).

## Tasks

Tasks included three categories: general cognitive tasks, numerical tasks, and parental home activity questionnaire.

**General cognitive tasks.**    *1. Matrix reasoning task (WPPSI IV)*. All participants aged 4 and older were administered the Matrix Reasoning Task from the Wechsler intelligence scales for children (Wechsler, D. & Psychological Corporation, 2012) [46] to assess their intellectual ability (the test is not suitable for younger children). Matrix reasoning is a non-verbal intelligence subtest. From visually presented response options, the child selects the one that best completes a matrix.

*2. Letter repetition*. Phonological loop capacity was measured by using a letter repetition task. Children listened to a series of one-syllable letter names recorded on the computer at the rate of one per second and were then asked to repeat them in the same order. The child's repetition was recorded for more accurate scoring. Only twelve letters with the same pronunciation in Vietnamese and French were chosen (i.e., B, C, G, K, L, M, N, P, R, S, T, V). A series of two letters were presented to begin the task, and the length of the series then progressively increased by one letter. Stimuli varied in length from one letter (e.g., K) up to five letters (e.g., P L G R V). Three items were built for each length, making a total of 15 items. After two failed trials of a given length, the task was stopped. The dependent variable corresponded to the higher length for which at least two trials were correctly repeated, plus .5 if one trial of the next length series was successful.

**Numerical tasks.**    *1. Counting*. This task was used to assess mastery of the counting sequence. The child was asked to count out loud as far as possible and was stopped when he/she reached 50. If required, a prompt (1, 2. . .) was given. If the child counts to 10 or makes an error in the counting sequence, the experimenter proposed a second trial. The score was the largest number without errors the child reached after two trials. The higher number words counted was used as the dependent variable.

*2. Counting on from a number (advanced counting)*. This task aims to assess the advanced level of counting–the elaboration of the counting sequence (e.g., [19]). Children were asked to *"count from N"* (N = 3, 5, 7, 13, 11, 12). If, for a trial (e.g., count from 2), the child did not start, the experimenter gave an example: *"I count from two, that is two, three, four, five, six."* The trial was successful if the child counted from the number required (and did not recite the previous number words or start from "one") and produced the next four numbers in the correct order. The task was stopped after three successive trial failures. The score was the total number of correct responses (CRs).

*3. Enumeration*. This task assessed the object counting skill and the cardinal principle of counting, i.e., that the last number words counted to represent the cardinal of the set (see [20] for details). A booklet showing a set of animals (N = 4, 6, 9, 10, 12, 15, 16, 17) arranged linearly and printed in color on 9 cm x 2.5 cm cards were placed in front of each child. They were invited to enumerate each set and then answer the question: *"How many animals are there*?*"* If they enumerated correctly and responded to the "*how many*" question with the last number words, the trial was considered successful, and the response coded 1. If they recounted or responded to the question by a number words different than the last one, the trial was considered incorrect, and these answers were coded 0. The total number of CRs were used as the dependent variable.

*4. Give-N (cardinal knowledge)*. This task was modeled on the one used by Negen and Sarnecka [47] to assess children's understanding of cardinal knowledge (i.e., the cardinal meaning of number words, see [22]). The experimenter placed a small bowl filled with plastic toys on a table in front of each child. They were asked to give the rabbit precisely 1, 2, 3, 4, 5, or 6 carrots from the bowl. *"Could you take one carrot out of the bowl and put it on the table for the rabbit*?*"* If a child did not react, the experimenter provided an illustrated demonstration. Children began at set size one and advanced to the next set size after a correct response but went down one set size after an incorrect response. The task stopped when a child made two mistakes on the same number or arrived at the larger number (i.e., give 6). This rule was applied when children were unsuccessful at any number (whether small or large). The child's knower-level corresponds to the highest number they reliably could give. For example, children who succeeded in "give one" and "give two" but failed at "give three" had a knower-level of two and were called *two-knower*. Children who had correctly handle the Give-N task for a number equal to five or above were called Cardinal Principle-knower (i.e., CP-knower). Children were thus classified as one-knower, two-knower, three-knower, four-knower, and CP-knower.

*5. Number-word comparison (NW-comparison)*. In this computerized task, developed by Honoré and Noël [48], children were asked to help Scrat have plenty of nuts by telling him which of two number words, given orally, is the largest. Stimuli were of two distances (close: distance of 1 and far: distance of 3) and two sizes (small: 1–9 and large: 11–19). There were 24 trials in total: 6 trials in each of the conditions (close distance—small size: 2–3, 5–6, 8–9, 8–7, 5–4, 5–3; far distance—small size: 2–5, 4–7, 6–9, 9–6, 8–5, 7–4; close distance—large size: 10–11, 13–14, 16–17, 18–17, 16–15, 19–18; and far distance—large size: 11–14, 15–18, 16–19, 19–16, 17–14, 18–15). All trials were presented in a fixed random order, according to four criteria: maximum three consecutive same-answer items, maximum two successive same-condition items, no following items of identical pair, and the first two items corresponded to far-small pairs. The total number of CRs was used as the dependent variable.

*6. Collection comparison*. This task was developed by Rousselle, Dembour, and Noël [49] and adapted by Honoré and Noël [48]. It aims to determine the child's approximate number sense (ANS) acuity. In this computerized test, participants were asked to help Dora find the most puzzle pieces. Children were presented with two boxes containing pieces of a puzzle (of various sizes) and asked to select the box containing the larger set of puzzle pieces. First, a fixation cross appeared in each box; once the child was attentive, the experimenter pressed the space bar, and the two collections were simultaneously displayed on both sides of the screen. To prevent children from counting, the collections disappeared after 2000 ms. The fixation cross appeared again with a question mark until the child answered. The perceptual variables were controlled to prevent children from relying on non-numerical parameters (see [50] for details). First, the external perimeter was equated for all collections. Second, both collections of a pair had the same smallest and the same largest puzzle piece. Finally, congruent and incongruent trials were built (half of each). In the congruent trials, the larger collection in number

also had the larger density and the larger cumulative area of the puzzle pieces, whereas, in the incongruent trials, the larger collection in number had the smaller density and the smaller cumulative area of the puzzle pieces. The number of puzzle pieces varied from 5 to 18, i.e., above the subitizing range (1−4), which represents the number of items for which humans are able to make fast and error-free estimations and which does not rely on analog numerical representation, hence is not related to the ANS (see [39] for a review). Six practice pairs of sets differing by a ratio of 1:3 familiarized children with the task. The test trials were pairs of sets differing by ratios of 1:2, 2:3, 3:4, 4:5, 5:6, 6:7, and 7:8. There were two pairs per ratio (7−14 and 8−16; 6−9 and 10−15; 6−8 and 12−16; 8−10 and 12−15; 5−6 and 10−12; 6−7 and 12−14; 7−8 and 14−16), and each pair was presented four times, varying according to the presentation order (smaller set on the left or the right side) and the condition (congruent or incongruent), resulting in 56 items. Items were presented in a fixed random order, respecting five criteria: maximum three consecutive same-answer items, maximum three consecutive same-condition items, maximum two consecutive same-ratio items, no consecutive items of identical pair, and the first two pairs were 1:2-ratio items. The total number of CRs was calculated and used as the dependent variable.

*7. Addition.* This task was similar to the one used by Noël [51]. A series of 14 additions that sum to 10 or less was given to each child: four ties (2 + 2, 3 + 3, 4 + 4, 5 + 5) and ten additions presented with the smaller addend first (1 + 3, 3 + 4, 2 + 3, 3 + 5, 1+ 4, 2 + 5, 2 + 4, 1 + 5, 4 + 5, 1 + 6) to allow a distinction to be made between the counting-on and the counting-min strategies. For each item, the child had a drawing of a collection of items (apples) representing the first operand, and tokens (10) and plastic apples (10) were at his or her disposal. The problem was presented orally (e.g., "*Look, here Snow White has three apples; if the dwarfs give her four more, how many apples will she have*?"). If the child failed three successive trials, the task was stopped; otherwise, the task was continued up to the last item. The total number of CRs was used as the dependent variable.

*8. Approximate addition.* To evaluate the children's ability to estimate the result of large non-symbolic additions, we used the task developed by Gilmore, McCarthy, and Spelke [24]. The children were invited to play a computer game. In this task, the quantities were represented using Arabic digits. Two characters appeared on the left and right sides of the screen, respectively. During practice trials, the experimenter explained the problem to the children: *"Babar has four candies"* (pointing at the bag displaying the Arabic number four). *"He gets six more"* (as a second bag displaying the Arabic number six appeared). Finally, a second character appeared on the right side of the screen with a bag showing an Arabic numeral (number six), and the experimenter said, *"Celeste has six candies. Who has more*?". The answer box was used with two sides: the left side corresponded to Babar, the right to Celeste. If Babar (Celeste) has more candies than Celeste (Babar), the child should press on the left (right). This task entailed two practice and 16 test trials. The 16 test trials corresponded to four additions (5+4, 7+5, 4+6, 6+5), corresponding to Babar's candies, associated with four other numbers, two smaller and two larger, representing Celeste's candies (5+4 and 6, 7, 11, 13; 7+5 and 8, 10, 15, 18; 4+6 and 7, 8, 12, 15; 6+5 and 8, 9, 14, 16). Items were presented in a fixed random order, respecting four criteria: maximum two consecutive same-ratio items; maximum two successive trials with the same answer; different additions for two consecutive trials; different numbers for Celeste's candies for two consecutive trials. The ratio between the sum (Babar's candies) and the last set (Celeste's candies) was 4/5 (e.g., 5+4/11) in half of the trials and 2/3 (i.e., 4+6/15) in the others. The total number of CRs was used as the dependent variable.

**Parental home activity questionnaire.** The parent questionnaire designed by Skwarchuk, Sowinski, and LeFevre [37] assesses home numeracy practices (13 items, e.g., counting, simple sums, printed number recognition) and home literacy practices (11 items, e.g., word reading,

written word recognition, sing/recite the alphabet). The parents (father or mother) answered by circling the corresponding frequency of each activity based on a Likert scale ranging from 0 (Never), 1 (Rarely), 2 (Sometimes), 3 (Often or once per week) to 4 (Very often or once per day). The mean frequency of home numeracy and home literacy practices was used as the dependent variable. The mean frequency of formal and informal activities were also analyzed.

## Procedure

The children were tested individually in a quiet room at their schools in two sessions of approximately 25 minutes for children who were 3½ and 45 minutes for children who were 4 years and older. The tasks were ordered, alternating between verbal and computerized tests, with a maximum of six tasks in a session.

The first session included counting, counting on from a number, enumeration, collection comparison, Give-N, and number-word comparison task. The second session included the matrix reasoning task (WPPSI), approximate addition, letter repetition, and addition.

Computerized tasks (collection comparison, approximate addition, and number-word comparison) were developed using E-prime experimental software (Version 2.0, Psychology Software Tools, Inc., Pittsburgh) and used a PC with a response box (left and right touch).

Children aged 3½ (N = 37; 17 VN, 20 BEL) completed seven tasks (exclusion: matrix, approximate addition task, addition) while children who were 4 years and older (N = 171; 87 VN, 84 BEL) completed all ten tasks. The parents were invited to complete a questionnaire about their child (when he/she started school, whether he/she has any developmental delay or diagnosed disorder) and a questionnaire about home numeracy and literacy practices and their highest education level. Measures employed were translated from the French version into Vietnamese by the first author and a Vietnamese doctor in psychology and then piloted on 40 Vietnamese and Belgian children. The official collection of data was carried out in April and May in Vietnam and October and November in Belgium.

## Results

### Analyses

The data were analyzed in four sections. In the first one, non-verbal intelligence and PL capacities across groups were compared. Non-verbal intelligence measured only in children aged 4 years and older was analyzed using a one-way ANOVA by nationality (two levels: Vietnam and Belgium) as a between-subjects factor. PL was introduced in an ANCOVA by nationality as a between-subjects factor and age (expressed in months) as a covariate.

In the second section, children's numerical performance was compared between the two countries using MANCOVA with age as the covariate. The first MANCOVA was run on the six numerical tasks that were presented to all children (i.e., counting, advanced counting, enumeration, Give-N, collection comparison, and number-word comparison). The second one was run on the performance of the two tasks that were administered only to children aged 4 years and older (i.e., the addition and approximate addition tasks). Further, we performed Bayesian analyses to compare the numerical performance of the Belgian and Vietnamese children. Gender was excluded from the analyses for two reasons. The first reason is based on the finding of a recent study which showed gender equality in the numerical competencies of 4- to 5-year-old preschoolers [52]. The second is based on our data's pilot analyses, which suggest no significant effects of gender and no interaction effect of gender x nationality (all $ps > .05$). We were unable to introduce the tasks' difficulty level (i.e., the ratio for collection comparison and approximate addition, size for enumeration, and number-word comparison) in the MANCOVA. However, previous analyses run separately on each task indicated that this did not

substantially change the results. Finally, a repeated measures ANCOVA was conducted to compare home activity frequency (numeracy and literacy) based on country and age (the covariate). Moderation analyses were used to investigate nationality's effect on the correlation between numerical abilities and home numeracy. Additionally, regression was carried to analyze the relationship between the children's counting ability and other factors such as nationality, age, and home numeracy.

## General cognitive ability

Non-verbal intelligence was tested on 171 children (only those aged 4 years and older). Standard scores were similar in the Vietnamese ($M = 9.65$, $SD = 2.87$) and Belgian groups ($M = 9.27$, $SD = 2.55$), $F(1, 169) = 0.73$, $p = 0.395$, $\eta^2 = 0.004$. The PL capacity was measured on all the 208 children. The ANCOVA (with age as a covariate) showed that PL capacity increased with age, $F(1, 205) = 14.14$ $p < 0.001$, $\eta^2 = 0.065$ but it was similar between Vietnamese ($M = 2.63$, $SD = 0.60$) and Belgian children ($M = 2.83$, $SD = 0.96$), $F(1, 205) = 2.84$, $p = 0.093$, $\eta^2 = 0.014$.

## Cross-national variation in numerical ability

The numerical performance of children in the two countries is presented in Table 2. Preliminary analyses run on computer tasks using a binary choice response showed that performance was above chance level (*t*-tests indicates that mean performance is significantly above 12/24 in number-word comparison (for VN: $t(103) = 6.73$, $p < 0.001$ and for BEL: $t(103) = 8.34$, $p < 0.001$) and above 28/56 in collection comparison (i.e., for VN: $t(103) = 8.16$, $p < 0.001$ and for BEL: $t(103) = 7.42$, $p < 0.001$) and above 8/16 in approximate addition, (i.e., for VN: $t(86) = 5.93$, $p < 0.001$ and for BEL: $t(83) = 2.27$, $p < 0.001$).

The first MANCOVA run on counting, advanced counting, enumeration, Give-N, collection comparison and number-word comparison showed a significant age effect (using Pillai's trace, $V = 0.42$, $F(6, 200) = 23.86$, $p < 0.001$, $\eta^2 = 0.417$) and nationality effects ($V = 0.23$, $F(6, 200) = 9.98$, $p < 0.001$, $\eta^2 = 0.230$). Performance increased with age in all tasks (counting, $F(1, 205) = 66.12$, $p < 0.001$, $\eta2 = 0.244$, counting from a number, $F(1, 205) = 34.16$, $p < 0.001$, $\eta^2 = 0.143$, enumeration, $F(1, 205) = 54.31$, $p < 0.001$, $\eta^2 = 0.209$, Give-N, $F(1, 205) = 98.23$, $p < 0.001$, $\eta^2 = 0.324$, number-word comparison, $F(1, 205) = 57.50$, $p < 0.001$, $\eta^2 = 0.219$, collection comparison, $F(1, 205) = 18.67$, $p < 0.001$, $\eta^2 = 0.084$). As regards the nationality effect (see Table 2), it was only significant in the counting task and marginally significant for the Give-N task. The second MANOVA run on addition and approximate addition, led to a significant age effect ($V = 0.19$, $F(2, 167) = 19.38$, $p < 0.001$, $\eta^2 = 0.19$) showing that performance

**Table 2. Comparison of numerical abilities between Vietnamese and Belgian children.**

| Abilities | Max | N | Mean ± SD | | Nationality effect |
|---|---|---|---|---|---|
| | | | Vietnam | Belgium | |
| Counting | 50 | 208 | 28.29 ± 14.04 | 19.72 ± 10.94 | $F(1, 205) = 35.18$, $p < 0.001$, $\eta^2 = 0.146$ |
| Advanced counting | 6 | 208 | 3.21 ± 2.51 | 3.51 ± 2.34 | $F(1, 205) = 0.54$, $p = 0.464$, $\eta^2 = 0.003$ |
| Enumeration | 8 | 208 | 3.81 ± 3.10 | 3.98 ± 2.87 | $F(1, 205) = 0.04$, $p = 0.849$, $\eta^2 = 0.000$ |
| Give-N | 6 | 208 | 4.52 ± 1.97 | 4.98 ± 1.55 | $F(1, 205) = 3.60$, $p = 0.059$, $\eta^2 = 0.017$ |
| NW-Comparison | 24 | 208 | 14.93 ± 4.44 | 15.14 ± 3.82 | $F(1, 205) = 0.14$, $p = 0.712$, $\eta^2 = 0.003$ |
| Collection comparison | 56 | 208 | 31.99 ± 4.98 | 31.58 ± 4.91 | $F(1, 205) = 0.67$, $p = 0.414$, $\eta^2 = 0.001$ |
| Addition | 14 | 171 | 4.26 ± 5.45 | 4.98 ± 5.21 | $F(1, 168) = 0.31$, $p = 0.580$, $\eta^2 = 0.002$ |
| Approximate addition | 16 | 171 | 9.75 ± 2.75 | 10.25 ± 2.85 | $F(1, 168) = 0.86$, $p = 0.354$, $\eta^2 = 0.005$ |

increased with age in both addition, $F(1, 168) = 20.99$, $p < 0.001$, $\eta^2 = 0.111$, and approximate addition, $F(1, 168) = 22.82$, $p < 0.001$, $\eta^2 = 0.120$. However, the nationality effect was not significant, $V = 0.01$, $F(2, 167) = 0.53$, $p = 0.592$, $\eta^2 = 0.006$.

Bayesian ANCOVAs with age as the covariate and nationality as the between-subjects factor was run on each numerical task (using JASP version 0.8.4). These analyses provide the Bayes factor (BF), which can be considered a relative measure of statistical evidence. According to Jeffreys [53], Bayes factor values of 1–3 are weak or inconclusive evidence, values of 3–10 are moderate evidence, values of 10–30 are strong evidence, values of 30–100 are robust, and values above 100 are extreme/decisive evidence for the presence of a given effect, including the null effect.

For counting, the best fitting model was the one with age and nationality effects ($BF_M =$ $2.99^e + 6$). Bayes Factor inclusion indicated that the data are $137^e + 11$ times more likely to occur in a model including the age effect and 766343 times more likely to occur under a model that includes the nationality effect than under the model without these effects. There is thus decisive evidence for both the age and nationality effects.

For the Give-N, the best fitting model is the one with age ($BF_M = 3.71$). The BF inclusion indicates that data are $9.007^e +15$ times more likely to occur in a model that includes the age effect. However, there is just weak evidence against the nationality effect. Indeed, data are only 1.23 times more likely to occur in a model that does not include this nationality effect.

For all the other numerical tasks, the best fitting model was the one with the age effect only ($BF_M = 15.45$ for advanced counting, $BF_M = 19.39$ for enumeration, $BF_M = 18.71$ for number-word comparison, $BF_M = 14.51$ for collection comparison, $BF_M = 15.685$ for addition and $BF_M = 12.11$ for approximate addition). The BF inclusion indicated that data were much more likely to occur in a model that included this age effect than under a model without it (644970.84 more likely for advanced counting, $2.00^e +9$ for enumeration, $7.022^e +9$ for number-word comparison, 682.45 for collection comparison, 2340.48 for addition, 5447.77 for approximate addition). There is also moderate evidence against the inclusion of a nationality effect. Indeed data are respectively 5.15 times for advanced counting, 6.45 times for enumeration, 6.25 times for number-word comparison, 4.87 times for collection comparison, 5.24 times for addition, 4.03 times for approximate addition more likely to occur in a model that does not include the nationality effect that in a model that does include it.

## Difference in counting ability and cardinal knowledge

To investigate deeper the nationality differences in counting, we considered four different levels of performance according to the higher category of number up to which the child could count: the unit level (one to ten), the teen level (eleven to sixteen), the small decade-unit numbers (seventeen to thirty) and the larger decade-unit numbers (thirty-one to fifty). Each child was categorized according to whether he/she had reached this level or not. Chi-square contingency tests were carried out to determine from what level any difference between Vietnamese and Belgian children appeared significant. The difference in counting ability was substantial from the teen's level, $X^2(1) = 27.56$, $p < 0.001$ (Fig 1). Thirty-one Belgian children, but only three Vietnamese children, stopped counting at sixteen or before and could not count further (see Table 3).

As regards the Give-N task, we considered the cumulated percentage of children in each sample, reaching each number-knower level, and chi-square contingency tests were calculated to examine from what number-knower level the difference between both samples appeared significant (see Table 4). The difference in number-knower level was significant at two-knower and indicated that more Belgian children could understand the cardinal meaning of "two" than Vietnamese children.

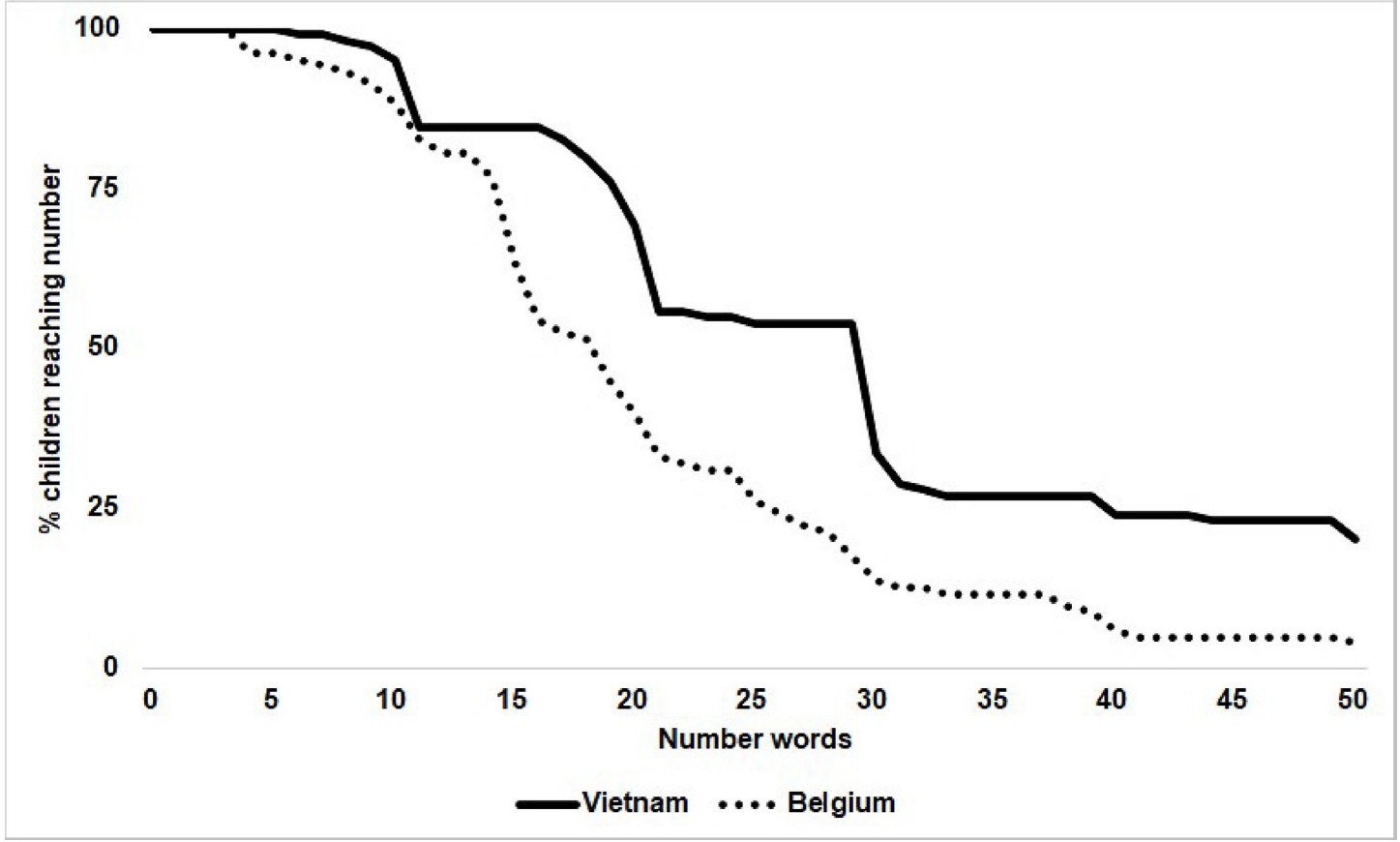

**Fig 1. Cumulated percentage of children able to count up to at least that number in the counting list.**

To sum up, we found an age effect in all the tasks, showing that all numerical abilities were sensitive to the child's development. More importantly, we found significantly better counting performance in Vietnamese children (starting at teens numbers) but marginally better numbers cardinal knowledge in Belgian children (especially regarding the numbers two). For all the other measures, the performance of the children in both countries was equivalent.

## Cross-national variation in home activity frequency

All parents completed the home activity questionnaire, but five Vietnamese parents did not answer all the questions. Cronbach's alphas were first computed on these data ($N = 203$) and showed good internal consistency for home numeracy (13 items, $\alpha = .86$) and home literacy (11 items, $\alpha = .89$). Scores for home numeracy and home literacy were calculated by averaging the score for all items, ranging from 0 (never) to 4 (very often or once per day).

**Table 3. Comparison of counting ability according to the counting sequence group.**

| Counting sequence Group | N Vietnam | N Belgium | X2(1) | p-value |
|---|---|---|---|---|
| One to ten | 15 | 19 | 0.56 | 0.453 |
| Eleven to sixteen | 3 | 31 | 27.56 | <0.001 |
| Seventeen to thirty | 56 | 41 | 4.34 | 0.037 |
| Thirty-one to fifty | 30 | 13 | 8.47 | 0.004 |

Table 4. Percentage of children reaching at least this number-knower level according to nationality.

| Number-knower level | Vietnam | Belgium | $X2(1)$ | $p$ |
|---|---|---|---|---|
| One-knower | 98 | 100 | 2.01 | 0.155 |
| Two-knower | 89 | 98 | 7.66 | 0.006 |
| Three-knower | 76 | 87 | 3.81 | 0.051 |
| Four-knower | 69 | 79 | 2.5 | 0.114 |
| CP-knower | 61 | 70 | 2.12 | 0.145 |

Each of these was entered into a repeated-measure ANCOVA with home activity (two levels: numeracy, literacy) as the within-subjects factor, and nationality as the between-subjects factor and age as a covariate. The effect of home activity was significant, $F(1, 200) = 4.92$, $p = 0.028$, $\eta^2 = 0.024$, indicating that the frequency of numeracy ($M = 2.25$, $SD = 0.68$) is higher than that of literacy ($M = 1.99$, $SD = 0.90$). The effect of age was also significant, showing that home numeracy frequency increased according to children's age. There was also a moderate main effect of nationality, showing that Vietnamese parents stimulated their child more frequently than Belgian parents, $F(1, 200) = 4.49$, $p = 0.035$, $\eta^2 = 0.022$ on both home numeracy ($M = 2.35$, $SD = 0.74$ vs. $M = 2.15$, $SD = 0.59$) and home literacy ($M = 2.11$, $SD = 1.02$ vs. $M = 1.88$, $SD = 0.75$, respectively). None of the interactions were significant (home activity x age, $F(1, 200) = 2.11$, $p = 0.147$, $\eta^2 = 0.010$; home activity x nationality, $F(1, 200) = 0.28$, $p = 0.599$, $\eta^2 = 0.001$).

## Difference in formal versus informal numeracy activity

According to the definition of formal and informal numeracy [37], we computed the two sub-scores of home numeracy, one for formal numeracy activity (i.e., the average score of six items 1, 2, 6, 8, 11, 13) and one for informal numeracy activity (i.e., the average score of seven items 3, 4, 5, 7, 9, 10, 12) (see Table 5).

Cronbach's alphas were computed on home numeracy data ($N = 203$) and showed acceptable internal consistency for formal numeracy subscale (6 items, $\alpha = .77$) and informal numeracy subscale (7 items, $\alpha = .76$). Independent sample $t$-tests were then used to compare the frequency of these two types of home numeracy activities (formal vs. informal) in the two samples. Vietnamese parents stimulated their child more on formal numeracy activities than Belgian parents, $t(184.16) = 2.33$, $p = 0.021$. However, the frequency of

Table 5. Frequency formal and informal home numeracy activities across nations.

| Items for formal numeracy activity | Items for informal numeracy activity |
|---|---|
| 1. I help my child learn simple sums (e.g., 2 + 2) | 3. We talk about time with clocks and calendars |
| 2. I encourage my child to do math in his/her head | 4. I help my child weigh, measure, and compare quantities |
| 6. I teach my child to recognize printed numbers | 5. We play games that involve counting, adding, or subtracting. |
| 8. I ask about quantities (e.g., How many candies?). | 7. We sort and classify by color, shape, and size. |
| 11. I help my child recite numbers in the sequence (1 2 3 4 5). | 9. We play board games or cards (dominoes, card games). |
| 13. I encourage the use of fingers to indicate how many | 10. I encourage collecting (e.g., cards, stamps, rocks). |
|  | 12. We sing counting songs (e.g., Five Little Fingers) |
| Mean ± SD for VN: 2.62 ± 0.78 | Mean ± SD for VN: 2.14 ± 0.81 |
| Mean ± SD for BEL: 2.39 ± 0.63 | Mean ± SD for BEL: 1.98 ± 0.68 |

informal numeracy activities did not differ between Vietnamese parents and Belgian parents, $t(186.88) = 1.55$, $p = 0.122$. Levene's test indicated unequal variances for formal numeracy activity, $F = 7.97$, $p = 0.005$, and for informal numeracy activity, $F = 4.48$, $p = 0.035$, so degrees of freedom were adjusted from 198 to 184.16 and from 195 to 186.88, respectively. However, according to Bayesian $t$-tests, the evidence in favor of a difference between the formal numeracy activities between Vietnamese and Belgian parents and the absence of such a difference for informal activities are weak (respectively, $BF_{10} = 1.28$ for formal activities and $BF_{01} = 1.96$ for informal ones).

## Home numeracy and children's numerical performance

Correlation between home numeracy frequency and numerical ability was calculated using nationality as a moderator. Children's performance in each numerical task (counting, advanced counting, enumeration, Give-N, collection comparison, and number-word comparison, addition, and the approximate addition task) was introduced in a Moderation running through Hayes' PROCESS (version 3.5) [54] by SPSS (version 25) with home numeracy as an independent variable and nationality (0: Belgium; 1: Vietnam) as a mediator.

Home numeracy correlated significantly with all precise symbolic numerical abilities (counting, advanced counting, enumeration, Give-N, number-word comparison, and addition), but not with the approximate numerical tasks (collection comparison and approximate addition) (See Table 6). These correlations were not affected by nationality. The effect of the moderator (nationality) and the interaction between home numeracy and nationality were no significant (all $ps > 0.05$).

## Language, home numeracy, and counting variation

Multiple regression was used to measure the weight of the different explanatory factors to account for the child's counting skills. In an initial analysis, age (expressed in months), home numeracy, and nationality (1: VN and 0: BEL) were entered into the model. The regression model explained 36.6% of the variance of counting performance, $F(3, 202) = 39.88$, $p < 0.001$. According to the *Beta* coefficients, age was the best explanatory factor ($\beta = .424$), followed by nationality ($\beta = .312$) and home numeracy ($\beta = .239$).

Subsequently, to see whether nationality still accounted for a significant part of the variance beyond age and home numeracy, hierarchical multiple regression was conducted. In the first block, age and home numeracy were forced, and then nationality was introduced in the second block using the stepwise method. The model (see Table 7), including age and home numeracy, was significant, $F(2, 200) = 38.9$, $p < 0.001$, and explained 28% of the variance in counting performance. However, nationality accounted for an additional significant 9.5% of the variance.

## Discussion

The main question addressed in this paper is: *"To what extent does a language's number-naming system impact preschoolers' numerical development?"* To disentangle the effect of cultural and language differences, we chose children from Vietnam (a former French colony) instead of another less culturally comparable Asian sample and compared them with children from Belgium's French-speaking region. Additionally, we assessed the home learning environment via a parent questionnaire. Furthermore, we tested preschool children to decrease the possible impact of differences in math school curricula. The two samples were very comparable in terms of age and gender, IQ and PL abilities, and parent education level.

**Table 6. Relationship between home numeracy and numerical performance using nationality as the moderator.**

|  | b | SE B | t | p | R² |
|---|---|---|---|---|---|
| *Counting* |  |  |  |  |  |
| Constant | 8.54 | 4.44 | 1.93 | 0.056 | 0.17 |
| Home Numeracy | 5.17 | 1.98 | 2.61 | **0.009** |  |
| Nationality | 4.81 | 5.98 | 0.80 | 0.422 |  |
| Home Numeracy x Nationality | 1.29 | 2.56 | 0.50 | 0.615 |  |
| *Advanced counting* |  |  |  |  |  |
| Constant | 1.19 | 0.87 | 1.37 | 0.172 | 0.27 |
| Home Numeracy | 1.07 | 0.39 | 2.77 | **0.006** |  |
| Nationality | -0.04 | 1.17 | -0.00 | 0.997 |  |
| Home Numeracy x Nationality | -0.18 | 0.50 | -0.37 | 0.711 |  |
| *Enumeration* |  |  |  |  |  |
| Constant | 1.79 | 1.08 | 1.65 | 0.100 | 0.22 |
| Home Numeracy | 1.01 | 0.48 | 2.09 | **0.037** |  |
| Nationality | -0.12 | 1.46 | -0.08 | 0.933 |  |
| Home Numeracy x Nationality | -0.10 | 0.62 | -0.16 | 0.869 |  |
| *Give-N* |  |  |  |  |  |
| Constant | 3.66 | 0.63 | 5.82 | < .001 | 0.27 |
| Home Numeracy | 0.61 | 0.28 | 2.17 | **0.031** |  |
| Nationality | -0.62 | 0.85 | -0.73 | 0.462 |  |
| Home Numeracy x Nationality | 0.05 | 0.36 | 0.13 | 0.667 |  |
| *NW-Comparison* |  |  |  |  |  |
| Constant | 10.23 | 2.70 | 3.78 | < .001 | 0.28 |
| Home Numeracy | 3.56 | 1.20 | 2.95 | **0.004** |  |
| Nationality | 0.78 | 3.64 | 0.21 | 0.830 |  |
| Home Numeracy x Nationality | -0.78 | 1.56 | -0.50 | 0.616 |  |
| *Collection comparison* |  |  |  |  |  |
| Constant | 28.47 | 1.80 | 15.79 | < .001 | 0.18 |
| Home Numeracy | 1.43 | 0.80 | 1.78 | 0.077 |  |
| Nationality | 0.72 | 2.42 | 0.29 | 0.767 |  |
| Home Numeracy x Nationality | -0.23 | 1.04 | -0.22 | 0.822 |  |
| *Addition* |  |  |  |  |  |
| Constant | 0.06 | 2.26 | 0.03 | 0.978 | 0.28 |
| Home Numeracy | 2.24 | 0.99 | 2.25 | **0.026** |  |
| Nationality | -0.94 | 2.96 | -0.32 | 0.750 |  |
| Home Numeracy x Nationality | -0.01 | 1.25 | -0.01 | 0.996 |  |
| *Approximate Addition* |  |  |  |  |  |
| Constant | 8.50 | 1.21 | 7.05 | < .001 | 0.17 |
| Home Numeracy | 0.81 | 0.53 | 1.52 | 0.131 |  |
| Nationality | 0.28 | 1.58 | 0.18 | 0.858 |  |
| Home Numeracy x Nationality | -0.42 | 0.67 | -0.62 | 0.532 |  |

## Differences in numerical ability between the two countries

Children's numerical development was tested through eight numerical tasks. Only one of these led to better performance in Vietnamese children compared to their French-speaking peers. Indeed, for simple rote counting, Vietnamese counted, on average, ten steps further than French-speaking children. This finding is consistent with that of previous studies that tested Chinese children [3, 6]. More specifically, we found that the differences between the two

**Table 7. Results of hierarchical multiple regression for counting performance.**

|              | B      | SE B | β    | t     | Sig. (p) |
|--------------|--------|------|------|-------|----------|
| Step 1       |        |      |      |       |          |
| Constant     | -33.40 | 6.87 |      | -4.86 | <0.001   |
| Age          | 0.81   | 0.12 | .407 | 6.72  | <0.001   |
| Numeracy     | 5.59   | 1.19 | .285 | 4.70  | <0.001   |
| Step 2       |        |      |      |       |          |
| Constant     | -37.31 | 6.46 |      | -5.78 | <0.001   |
| Age          | 0.85   | 0.11 | .424 | 7.49  | <0.001   |
| Numeracy     | 4.69   | 1.12 | .239 | 4.17  | <0.001   |
| Nationality  | 8.32   | 1.51 | .312 | 5.52  | <0.001   |

*Note*: $R^2 = .280$ for step 1, $R^2 = .375$ for step 2; $\Delta R^2 = .095$ for step 2.

samples started to emerge at the level of the teen numbers; that is, in an area where French-speaking children have to learn specific words corresponding to the teens, whereas Vietnamese children follow regular rules, combining the words for ten and the words for the unit. This result is strikingly similar to the findings of a previous study [3] and provides additional evidence that a transparent, regular number-naming system facilitates learning of counting sequences in preschoolers.

In contrast, the transparent, regular Vietnamese number-naming system did not facilitate other numerical tasks. More specifically, when we used a more advanced counting task where children had to start counting from a number different from one, which assesses the level of the *"breakable chain"* (e.g., see [33]), we failed to identify any advantage for the Vietnamese children. Thus, although the Vietnamese children's counting chain was longer than that of the Belgian children by approximately ten numbers, they did not develop faster regarding the elaboration of their number sequences. These results are not in line with those of a previous study [4], but the authors of that work compared Chinese children with British and Finnish children; the former might have received more educational stimulation.

Regarding enumeration (i.e., using the numerical chain to determine the number of items in a set), unlike the previous study [3], no overall difference was found. We should remember that object counting requires the ability to produce a counting sequence in the right order, a certain level of elaboration of the counting sequence (i.e., the level of the unbreakable chain, see [19]), and mastery of different counting principles (e.g., see [20]). Thus, although the Vietnamese children's counting sequence was higher, they did not perform better on the more complex collection counting task.

We also used two tasks to test the cardinal understanding of number words: the Give-N task and the number-word comparison task. Neither of these demonstrated an advantage for the Vietnamese children. In contrast, we found that French-speaking children had marginally better cardinality knowledge than Vietnamese children. These results are particularly impressive given that a previous longitudinal study [55] tested a large sample of preschoolers and found that those who knew more count words at the beginning of preschool also performed better on the Give-N task. Conversely, the same research team [56] found that a good understanding of number words' cardinal meaning was a strong predictor of counting abilities two years later when the children were six years old. Thus, although these studies show a bidirectional link between number-word cardinal understanding and count list extension, our research indicates that these two abilities may develop somewhat differently in young Vietnamese children since despite their count list being longer than French-speaking peers, they

lagged behind in terms of understanding number words' cardinal meaning (especially learning the number *"two"*). Therefore, we support the previous view of Wynn [22], who showed that the acquisition of the cardinal meaning of number words does not coincide with the sequence counting ability or object counting ability.

Interestingly, this result is consistent with that of previous studies showing that Asian children (i.e., Japanese, Chinese) were not better but were slower in understanding the cardinal meaning of the number words *"one," "two,"* and *"three"* than Western children (i.e., English and Russian) [31, 32]. Vietnamese is like Japanese and Chinese, where there is no distinct singular/plural, whereas this distinction is obligatory in French and English. In the current study, we found that French-speaking children outperformed Vietnamese children in understanding the cardinal value of the numbers *"two,"* but the differences by country disappeared regarding learning other numbers. This finding suggests that the singular-plural distinction of language, but not the transparency of the number-naming system, does influence cardinality knowledge of the small number words *"two."*

The number-word comparison task was never used in previous studies comparing Asian and Western preschoolers. Here, again, we observed no difference in performance between our two samples. Thus, although the counting sequence is larger in Vietnamese children than in French-speaking children, it was not associated with a better ability to compare the magnitude of larger number words. Recently, Sella, Lucangeli, and Zorzi [57] showed that the comparison of number words (i.e., below 9) is related to cardinality knowledge measured using the Give-N task. Accordingly, we propose that in comparing the magnitude of two number words, it is important not only to know the number words' sequence but also to understand their cardinal meaning.

For simple addition, we found no difference between the two samples. This finding is inconsistent with that of the previous study, which showed the large effect size of nationality effect (Cohen's $f$ = 1.03 [7]), supporting Chinese children's higher performance. However, the current study sample was younger than Chinese kindergarteners ($M_{age}$ = 71 months). We suggested that the absence of a group effect was not due to the sample size ($n$ = 51 in [7] vs. $n$ = 171 in our study). Furthermore, most researchers have reported that Chinese kindergarteners engage in regular mathematics learning sessions [10], which is not the case in Vietnam and Belgium (i.e., without arithmetic instruction in preschool). In addition, we excluded Vietnamese preschoolers who attended extracurricular math programs from the study.

We also wanted to examine whether Asian children's possible advantage in symbolic number processing tasks also extended to tasks that do not involve precise number processing but simply tap into the processing of approximate magnitude. To that end, we used both a collection comparison task and an approximate addition task. Here, again, the two samples behaved similarly, thus supporting the view that at that age, language transparency does not influence non-symbolic ability. Our finding is inconsistent with those of previous studies [5, 6], which found that Chinese preschoolers were better at non-symbolic magnitude comparison than other groups. However, our results are consistent with those of a recent study [58], which showed that 6- to 9-year-old Chinese children were faster but had slightly lower accuracy than Russian and British children in symbolic and non-symbolic number magnitude comparisons.

One possible explanation for these inconsistent results could be associated with differences in symbolic number tasks such as arithmetic and symbolic number comparisons. Indeed, according to the refinement hypothesis proposed by Noël and Rousselle [59], more frequent and advanced practices with exact (symbolic) numbers would lead to a refinement of the approximate number system. Similarly, Chen and Li [60] concluded their meta-analysis by suggesting a bidirectional relationship between non-symbolic magnitude processing and symbolic mathematics performance, each influencing the other. In our study, since we did not find

any advantage in Vietnamese children regarding symbolic tasks such as number-word comparison or addition, it is not surprising that there was no difference in the non-symbolic tasks either. The better achievement in Chinese children's non-symbolic numerical tasks can be attributed to early education in school (unlike other countries in Asia or Europe).

## Differences in home numeracy by country

Asian parents and society place a strong emphasis on mathematics and have high expectations in this regard [10]. To consider this point, we measured the influence of parent-child numerical stimulation on children's numerical development. We found that Vietnamese parents generally stimulate their child slightly more than Belgian parents in numeracy and literacy. Furthermore, we found that the kind of home numeracy slightly differed across the groups. The Vietnamese parents tended to practice more frequent formal numeracy activities (i.e., simple arithmetic, mental addition) with their children than the Belgian parents, whereas there was no difference in the frequency of informal numeracy activities. These findings are congruent with those reported previously and show that Asian parents use more formal instruction related to arithmetic at home [13, 14].

Nevertheless, despite this more frequent numeracy stimulation (especially regarding formal numeracy), the Vietnamese children did not exhibit more advanced symbolic numerical development, except for basic counting. However, the average frequency reported for home numeracy was approximately 2 for the Vietnamese parents, which roughly corresponds to *"sometimes,"* a frequency that might be too low to lead to better addition performance in this group than the Belgian children. Beyond the frequency of home numeracy, other factors should play a role, such as the quality of the practices and the kinds of numeracy activities [35].

Finally, for the sole task where Vietnamese children performed better than Belgian children (the counting task), we looked at the explanatory power of the language's effect (expressed by nationality) and the impact of home experience (expressed by home numeracy). The regression analysis indicated that both factors played a significant role in explaining the counting skills. These results expand on the findings of Cankaya, LeFevre, and Dunbar [38], who showed that both the frequency of parent-child numeracy stimulation and the number-naming system impacted counting skills.

## Strengths and limitations

Our research has several strengths. We assessed preschoolers from Vietnam and Belgium, thus reducing the difference in mathematics school instruction. The two samples were equivalent in several aspects (age and parent education level, except for a moderate difference in formal home numeracy). Thus, the two groups were very similar, allowing us to measure the number-naming systems' influence on children's numerical development more selectively. Furthermore, we assessed the children's numerical development with eight different tasks, whereas most previous studies on this topic have had at most three (e.g., counting, enumeration, and addition). Finally, we also controlled for parents' stimulation of the child as a possible contributory factor in their numerical development. Such an approach was barely used in previous studies examining the impact of the number-naming system, except for that of Cankaya, LeFevre, and Dunbar [39].

However, our study also has some limitations. First, although we chose the location (large cities) and parent education level as two criteria to select comparable samples, we failed to measure a parent's socioeconomic level. However, Davis-Kean [61] found direct effects of parent education (but not income) on European American children's standardized achievement scores. Furthermore, the incomes could not be directly compared, as the cost of living is very

different between the two countries. Second, the home numeracy questionnaire allowed us to examine the home learning environment as one aspect of culture. However, we did not measure the parents' attitudes toward academic success. Future studies should elaborate on this factor. Third, in the Give-N task, we found some cases of children ($n$ = 7) who could give five objects but failed to give six, showing that being a CP-knower (i.e., Cardinal Principle knower) is not equivalent to being a four-knower or even a five-knower. A recent article has also reported on this issue [62]. Future research should extend the range of numbers used in the task (and include even numbers larger than 10) to precisely examine when children generalize their cardinality knowledge for all number words. This could be after understanding the number *"four"* for some children, the number *"five"* for others, and perhaps the number *"six"* or even a larger number for others.

Finally, regarding the effect size of nationality on numerical abilities, we found a moderate effect size on counting ($\eta2$ = 0.146), which is consistent with that of previous studies [3, 6]. However, we found a minimal nationality effect size ($\eta2$ = 0.017) on cardinal knowledge compared to the large effect size on the literature (Cohen's $f$ = 0.82, [31]). This could be explained because our sample was older than that in the previous study ($M_{age}$ = 3;2) and did not correspond to the critical age where children learn the cardinal meaning of their first number words. Future studies need to examine this ability in younger children to favor the comparison.

In short, we found that the transparency of the number-naming system in Vietnamese did not have an all-pervasive influence on the preschoolers' numerical performance. Thus, our results confirm the effect of the transparency of the number-naming system on counting in Vietnamese, but fail to find other advantages. By contrast, French-speaking children were more advanced in understanding the cardinal meaning of the number *"two,"* which we interpreted as being due to the numerical morphology (singular/plural distinction) in French but not in Vietnamese. Vietnamese parents stimulated their child's numerical development (mainly through formal home numeracy) slightly more than Belgian parents. However, both language and home numeracy effects accounted for counting variation between the two groups.

To conclude, we provided evidence that the Vietnamese number-naming system's transparency led to a faster acquisition of basic counting for preschoolers but did not support other more advanced numerical development or non-symbolic numerical abilities. In addition, we extended the evidence that both transparent number-naming system and home numeracy influence young children's counting development.

## Supporting information

**S1 Raw image.**
(PDF)

**S1 Data. Counting data.**
(XLSX)

**S2 Data. Advanced counting data.**
(XLSX)

**S3 Data. Enumeration data.**
(XLSX)

**S4 Data. Give-N data.**
(XLSX)

**S5 Data. Number-word comparison data.**
(XLSX)

**S6 Data. Collection comparison data.**
(XLSX)

**S7 Data. Addition data.**
(XLSX)

**S8 Data. Approximate addition data.**
(XLSX)

**S9 Data. Nationality, Age, IQ, phonological loop data.**
(XLSX)

**S10 Data. Parental education and home activity data.**
(XLSX)

## Acknowledgments

We thank all the children, parents, teachers, and principals in Vietnam and Belgium to partici-pate in this research. We also thank students of USSH (To Thi Mai Dao, Le Thanh Nhan, Nguyen Bao An, Nguyen Thao Nguyen, Dang Thi Thu Hang) and UCLouvain (Marie Clerb-out, Anne Van Eyll, Marie Demoulin, Justine Vandeuren) for collecting the data, Pierre Mahau and Phan Hoang Loc for software support, and Nastasya Honoré, Nathalie Lefèvre, Jacques Grégoire and Laurence Rousselle for their shrewd suggestions.

## Author Contributions

**Conceptualization:** Mai-Liên T. Lê, Marie-Pascale Noël.

**Data curation:** Mai-Liên T. Lê.

**Formal analysis:** Mai-Liên T. Lê.

**Funding acquisition:** Mai-Liên T. Lê, Marie-Pascale Noël.

**Investigation:** Mai-Liên T. Lê, Marie-Pascale Noël.

**Methodology:** Mai-Liên T. Lê, Marie-Pascale Noël.

**Project administration:** Mai-Liên T. Lê.

**Software:** Mai-Liên T. Lê.

**Supervision:** Marie-Pascale Noël.

**Visualization:** Mai-Liên T. Lê.

**Writing – original draft:** Mai-Liên T. Lê.

**Writing – review & editing:** Marie-Pascale Noël.

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
