## [Decision Letter · Decision Letter 0]

1 Jul 2020

PONE-D-20-10381

The Impact of Language on Numerical Development: Dissociation between Counting and Cardinality in Vietnamese and French-Speaking Preschoolers

PLOS ONE

Dear Dr. Lê,

Thank you for submitting your manuscript to PLOS ONE. I have sent your manuscript to two expert reviewers and have now received their comments back. As you can see attached and below, both reviewers are positive about your study and highlight the uniqueness and interesting aspect of your sample. I agree with them. Nonetheless, both reviewers also point to several important issues that I think you would need to address in a major revision. I will not reiterate all of the reviewers' points here, as the reviewers made them very clear. But I concur with the reviewers that you run a large number of analyses in the manuscript, which raises a multiple comparison issue that might be addressed with adequate corrections or a change in analysis strategy (see suggestions from reviewer #2). Both reviewers also highlight that Bayesian analyses might help you make the most of null results and I also think that this might be very informative. Therefore, I encourage you to address these issues (as well as all of the other concerns from the reviewers) and submit a revised version of your manuscript. I will send this version back to the original reviewers.

We look forward to receiving your revised manuscript.

Kind regards,

Jérôme Prado

Academic Editor

PLOS ONE

Journal Requirements:

2. Please improving statistical reporting and refer to p-values as "p<.001" instead of "p=.000". Our statistical reporting guidelines are available at https://journals.plos.org/plosone/s/submission-guidelines#loc-statistical-reporting.

3. Please upload a copy of all of the Supporting Information files (only one is uploaded, but 10 listed) which you refer to in your text on page 46.

Reviewers' comments:

Reviewer #1: 1. I would ask the authors to tone down their main conclusion a bit/explain in more depth why they think their data support their main conclusion.

2. I am not sure if the authors corrected for multiple comparisons (might change results partly)

3. Aggregated data is available for all tasks, item level data is missing for some of the tasks.

4. Nicely written, clear structure

Please see attachment for further comments.

Reviewer #2: Review of PONE-D-20-10381: The Impact of Language on Numerical Development: Dissociation between Counting and Cardinality in Vietnamese and French-Speaking Preschoolers

In this manuscript, the authors present cross-cultural data that reveals that Vietnamese children outperformed Belgian children in counting, but not in other symbolic and nonsymbolic tasks, with Belgian children actually outperforming Vietnamese children in cardinality knowledge. Additionally, this research reports that Vietnamese children tend to be more stimulated in the home environment. First, I wish to note that this is a very impressive dataset and I applaud the authors for collecting such a large cross-cultural sample. I think that this is an interesting and important research question and has the potential to add to the current state of our understanding of the influence of language on number processing in young children. However, I have many concerns regarding the methodological and statistical choices as well as several concerns about the theoretical setup and interpretation of this data. Below I outline both major and minor concerns.

Major Issues

Methodological

1. This manuscript contains many different analyses (my estimate: 12 ANOVAS, 10 Chi-squared tests, 13 t-tests, 24 correlations, 2 regressions). However, I don’t see any correction for multiple comparisons. I recommend reducing the number of analyses included where possible and including fewer, but more sophisticated analyses. I recommend analysing the cognitive and numerical data using a MANOVA, GLM or perhaps a mixed effects model, instead of multiple univariate ANOVAs. Similarly, rather than running correlation analyses on the full sample and each country separately to examine the relation between numerical ability and home numeracy, I recommend running moderation analyses. In the cases where multiple analyses are necessary, corrections for multiple comparisons should be included.

2. I have concerns about the decision to use median splits for continuous data, particularly with age (but also education level). There are statistical issues with converting continuous data into categories in general (see: https://www.theanalysisfactor.com/continuous-and-categorical-variables-the-trouble-with-median-splits/). It is particularly problematic for age given the dramatic changes observed in basic number processing during the age range included in the sample (e.g. Le Corre & Carey, 2007; Merkley & Ansari, 2016). The authors should use one of the analyses methods suggested above, with age included as a continuous predictor.

3. Many of the key conclusions drawn from this data are a result of null findings (i.e. that Belgian and Vietnamese children do not differ). Critically, the frequentist-based statistics used in the current study can only be used to reject the null hypothesis. They cannot be used to confirm that there is in fact no difference between groups. Therefore, in order to draw key conclusions (e.g. “the performance of the children in both countries with equivalent” – line 659), the authors should conduct and include follow-up Bayesian analyses to test if the null hypothesis is true. In line with this, in order to conclude that nationality does indeed improve the hierarchical model, a model comparison method (again Bayesian would do the trick here), is needed to compare the models.

Theoretical

4. The authors indicate that Vietnamese parent’s attitudes towards academic success are more similar to the European view, making French-Belgium and Vietnamese’s children ideal groups to compare. However, these statements are not supported by empirical evidence. I think the introduction would benefit from a stronger theoretical setup with supporting citations explaining why these two groups are ideal to compare.

5. Relatedly, I have several questions about the comparability of the samples.

a. Specifically, the authors indicate that the two groups have equivalent SES. This is based on years of education. My question is whether equivalent years of education results in similar occupational levels in both cultures? Additionally, as the samples are selected to be equivalent from larger samples, I am curious about where the Belgian and Vietnamese sample each fall within the context of the larger sample. More specifically, while the years of education between samples is reported as approximately equivalent, it is possible that one sample has low SES compared to the larger within country sample and the other has high SES compared to the larger within country sample. A difference in the relative position within the broader community should be checked for equivalence and potentially discussed.

b. Is it typical for individuals in each country to be monolingual? For example, is it common for children in Belgium to only speak French or does this community typically speak French and Belgian? Is there a difference in the degree to which children in each country are typically exposed to other languages?

c. I think that the authors made a great decision to test children at the preschool age, prior to formal schooling, to reduce the potential influence of differences in the educational system! The authors note that children who had additional math tutoring were excluded, but I did not see a report of the number of children enrolled in some sort of pre-school or daycare. I think the number of children enrolled in preschool/daycare should be reported. If many children are enrolled in pre-school, it could be important to consider whether each country has a preschool curriculum that is followed and how they differ.

Minor Issues

Theoretical

6. I would appreciate a more comprehensive overview of the process of acquiring symbolic number knowledge (e.g. the process of learning how to count; acquiring cardinal knowledge; distinction between procedural and conceptual knowledge).

7. The authors report one study (ref 31) where symbolic and nonsymbolic number processing influence each other. However, as this is a widely studied and debated topic in the field of numerical cognition. Since symbolic and nonsymbolic numerical knowledge are included and compared as variables of interest, I think the introduction could benefit from a slightly more comprehensive description of this area. I think this could also be helpful for setting up clearer predictions for symbolic and nonsymbolic number processing.

8. In the discussion, the authors note that there are differences in the kinds of home numeracy done in each country (formal vs informal). I thought this was an interesting and wonder if it should be included as an exploratory section in the results? I don’t feel strongly about this and leave it up to the authors!

Methodological

9. While the sample size seems sufficient to detect at least a medium effect size within the analyses included (and is impressive!), it would be helpful to include a power analysis to indicate what exact effect sizes can be detected with each analysis using the current sample.

10. The decision to include size as a within-subjects variable, grouped into two levels, needs further justification.

11. The Huynh-Feldt correction for sphericity is the least conservative correction and often considered to be too liberal and consequently overestimates sphericity. I am curious why the authors did not choose the more typical Greenhouse-Geisser? Unless epsilon is >.75, I recommend the authors switch to Greenhouse-Geisser (Girden, 1992).

12. A more complete description of the give me a number task would be helpful. For example, did the child need to correctly give-N twice at one number and incorrectly give-N twice at the number above to be considered that knower-level? Generally, did the authors use the traditional protocol (from Wynn, 1992)? If so, it would be best to cite the original paper. Also, I think the give me a number task is most typically referred to as the “Give-A-Number Task” or “Give-N”, rather than Give me a number. This isn’t super important but using the most common name might make it easier to find this article for those looking at this task.

13. As mentioned above, I do not follow the logic to conduct a median split on the education data and use a chi-squared test to compare years of education. Why did the authors choose this method rather than to keep years of education as a continuous variable and then run a t-test to compare years of education?

Stylistic

14. The figures are nice, but it would really great to see the data points plotted under the line (or violin plots) to understand not only the mean but also variability within the two samples. I leave this up to the authors.

15. The reporting of nonsignificant findings is inconsistent. Sometimes it is reported Is the f<1 and other times the nonsignificant pvalue is reported. I think the authors should stick to one of the two reporting styles.

16. The whole manuscript would benefit from a thorough proofread for grammar and clarity. In particular, there is a lot of phrasing that seems a bit odd and switching between tenses. A few examples of phrases that would benefit from editing include:

o “in the same way” – Line 137

o “Seems to be quite a significant difference” Line 160

o “The prediction… is still an open question” Lines 220-221

o “According to Chen and Li (2014), there would be a…” – Line 222

o “Belgian children performed not significantly better” – Line 543

o “There was a small interaction…” – Line 656

6. PLOS authors have the option to publish the peer review history of their article (what does this mean?). If published, this will include your full peer review and any attached files.

Reviewer #1: **Yes: **Julia Bahnmueller

Reviewer #2: No

---

## [Author Response · Author response to Decision Letter 0]

15 Aug 2020

Reviewer 1: I have incorporated all of your suggestions into my revision. They were precious! Thank you so much!

Reviewer 2: I have incorporated all of your suggestions into my revision. They were very helpful! Thank you so much!

---

## [Decision Letter · Decision Letter 1]

1 Oct 2020

PONE-D-20-10381R1

Transparent number-naming system gives only limited advantage for preschooler’s numerical development: Comparisons of Vietnamese and French-speaking children

PLOS ONE

Dear Dr. Lê,

Thank you for submitting your revised manuscript to PLOS ONE. I have sent it to the original reviewers and have received their comments back. As you can see below and attached, both reviewers found that the paper considerably improved. I agree this them. However, they have some lingering concerns that you would need to address before the manuscript can be accepted for publication in PLOS ONE. Notably, both reviewers are concerned about the post-hoc power analyses that you now provide. I actually agree with reviewer #2 that post-hoc power analyses are not very helpful and could be misleading because they do not reflect the true power of a study (simply the observed power, which is inversely related to the p value). Therefore, you should remove these analyses and either replace them with power analyses based on previous estimates of relevant effect sizes in the literature (e.g., coming from meta-analyses or previous studies) or discuss the sample size limitation in the discussion. I also encourage you to address the other reviewers' suggestions, which might improve the clarity of the manuscript.

We look forward to receiving your revised manuscript.

Kind regards,

Jérôme Prado

Academic Editor

PLOS ONE

Reviewers' comments:

Reviewer #1: 3. In my view the main analyses are fine, I am not sure about the post-hoc power analyses the authors presented

5.The manuscript needs to be proof read by a native English speaker; there are many grammatical errors and flaws in the manuscript. Also, the manuscript would profit from some further editing (style, explicitness) to guide the reader more through the quite longish paper.

Reviewer #2: The authors have made many of my recommended changes (all of the major ones) and I think this manuscript is much improved and of great value to the scientific community! I just have a couple remaining concerns:

1. The authors provide comprehensive explanation for why the two groups are ideal to compare in response to my Theoretical comment #4. I found this explanation quite helpful and therefore recommend that a condensed version of it be added into the manuscript.

2. Based on my (limited) understanding of power analyses, I believe that post-hoc power analyses are not typically used or considered useful (see: http://daniellakens.blogspot.com/2014/12/observed-power-and-what-to-do-if-your.html#:~:text=Observed%20power%20(or%20post%2Dhoc,size%20estimate%20from%20your%20data.&text=Observed%20power%20is%20a%20useless,for%20post%2Dhoc%20power%20analyses.) Therefore, I recommend removing these from the manuscript and instead just including a brief note of this in the limitation section.

---

## [Author Response · Author response to Decision Letter 1]

3 Nov 2020

We thank the Editor and the Reviewers for useful suggestions. We incorporated all of your suggestions into the final manuscript.

---

## [Editor Report · Decision Letter 2]

23 Nov 2020

Transparent number-naming system gives only limited advantage for preschooler’s numerical development: Comparisons of Vietnamese and French-speaking children

PONE-D-20-10381R2

Dear Dr. Lê,

I am pleased to inform you that your manuscript has been judged scientifically suitable for publication and will be formally accepted for publication once it meets all outstanding technical requirements.

Kind regards,

Jérôme Prado

Academic Editor

PLOS ONE

---

## [Editor Report · Acceptance letter]

26 Nov 2020

PONE-D-20-10381R2 

Transparent number-naming system gives only limited advantage for preschooler's numerical development: Comparisons of Vietnamese and French-speaking children 

Dear Dr. Lê:

I'm pleased to inform you that your manuscript has been deemed suitable for publication in PLOS ONE. Congratulations! Your manuscript is now with our production department. 

Kind regards, 

on behalf of

Dr. Jérôme Prado 

Academic Editor

PLOS ONE